# Language and Visual Entity Relationship Graph for Agent Navigation

**Yicong Hong**[1]    **Cristian Rodriguez-Opazo**[1]    **Yuankai Qi**[2]    **Qi Wu**[2]    **Stephen Gould**[1]

[1]Australian National University    [2]The University of Adelaide
[1,2]Australian Centre for Robotic Vision

{yicong.hong, cristian.rodriguez, stephen.gould}@anu.edu.au
qykshr@gmail.com, qi.wu01@adelaide.edu.au

## Abstract

Vision-and-Language Navigation (VLN) requires an agent to navigate in a real-world environment following natural language instructions. From both the textual and visual perspectives, we find that the relationships among the scene, its objects, and directional clues are essential for the agent to interpret complex instructions and correctly perceive the environment. To capture and utilize the relationships, we propose a novel *Language and Visual Entity Relationship Graph* for modelling the inter-modal relationships between text and vision, and the intra-modal relationships among visual entities. We propose a message passing algorithm for propagating information between language elements and visual entities in the graph, which we then combine to determine the next action to take. Experiments show that by taking advantage of the relationships we are able to improve over state-of-the-art. On the Room-to-Room (R2R) benchmark, our method achieves the new best performance on the test unseen split with success rate weighted by path length (SPL) of 52%. On the Room-for-Room (R4R) dataset, our method significantly improves the previous best from 13% to 34% on the success weighted by normalized dynamic time warping (SDTW).

Code is available at: https://github.com/YicongHong/Entity-Graph-VLN.

## 1    Introduction

Vision-and-language navigation in the real-world is an important step towards building mobile agents that perceive their environments and complete specific tasks following human instructions. A great variety of scenarios have been set up for relevant research, such as long range indoor and street view navigation with comprehensive instructions [3, 5, 17], communication based visual navigation [7, 25, 34], and navigation for object localization and visual question answering [6, 29].

The recently proposed R2R navigation task by Anderson et al. [3] has drawn significant research interest. Here an agent needs to navigate in an unseen photo-realistic environment following a natural language instruction, such as "*Walk toward the white patio table and chairs and go into the house through the glass sliding doors. Pass the grey couches and go into the kitchen. Wait by the toaster.*". This task is particularly challenging as the agent needs to learn the step-wise correspondence between complex visual clues and the natural language instruction without any explicit intermediate supervision (e.g., matching between sub-instructions and path-segments [12, 41]).

Most previous agents proposed for the R2R navigation task are based on a sequence-to-sequence network [3] with grounding between vision and language [9, 21, 22, 32, 36]. Instead of explicitly modelling the relationship between visual features and the orientation of the agent, these methods

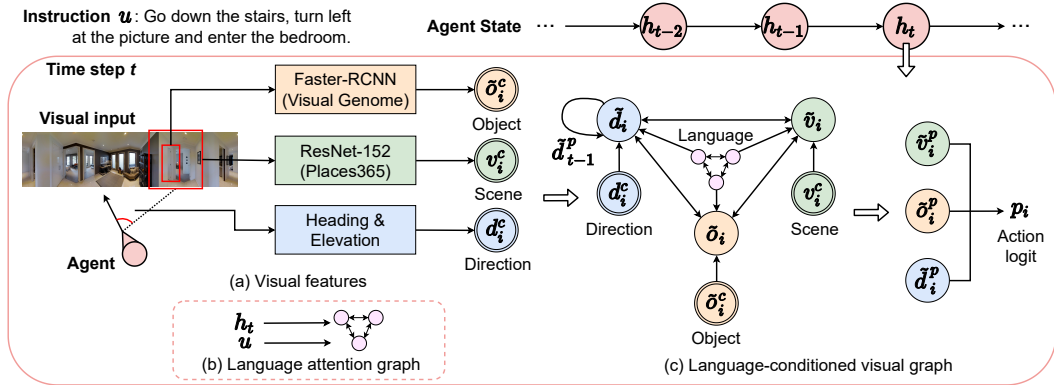

Figure 1: Language and Visual Entity Relationship Graph. At each navigational step $t$, (a) Scene, object and directional clues are observed and encoded as visual features. (b) language attention graph is constructed depending on the agent's state, (c) visual features are initialized as nodes in the language-conditioned visual graph, information propagated through the graph updates the nodes, which are ultimately used for determining action probabilities. Each double-circle in the figure indicates an observed feature.

resort to a high-dimensional representation that concatenates image features and directional encoding. Moreover, objects seen during navigation, which are mentioned in the instruction and contain strong localization signals, are rarely considered in previous works.

However, we observe that many navigation instructions contain three different contextual clues, each of which corresponds to a distinct visual feature that are essential for the interpretation of the instruction: scene ("*where is the agent at a coarse level?*"), object ("*what is the pose of the agent with respect to this landmark?*") and direction ("*what action should the agent take relative to its orientation?*"). Meanwhile, these contextual clues are not independent, but work together to clarify the instruction. For instance, given an instruction "*With the sofa on your left, turn right into the bathroom and stop*", the agent first needs to identify the "*sofa*" (object) as a landmark on its left, which the impending action is conditioned on, then "*turn right*" (direction) to move to the "*bathroom*" (scene), and finally "*stop*" (direction) inside the "*bathroom*" (scene). As a result, it is important for the agent to learn about the relationships among the scene, the object and the direction.

In this paper, we propose a novel language and visual entity relationship graph for vision-and-language navigation that explicitly models the inter- and intra-modality relationships among the scene, the object, and the directional clues (Figure 1). The graph is composed of two interacting subgraphs. The first subgraph, a language attention graph, is responsible for attending to specific words in the instruction and their relationships. The second subgraph, a language-conditioned visual graph, is constructed for each navigable viewpoint and has nodes representing distinct visual features specialized for scene, object and direction. Information propagated through the graph updates the nodes, which are ultimately used for determining action probabilities.

The performance of our agent, trained end-to-end from scratch on the R2R [3] benchmark, significantly outperforms the baseline model [32] and achieves new state-of-the-art results on the test unseen split following the single-run setting[1]. On the R4R data [17], an extended dataset of R2R, our method obtains 21% absolute improvement on the Success weighted by normalized Dynamic Time Warping [15] comparing to the previous best.

We believe the proposed language and visual entity relationship graph will prove valuable for other vision-and-language tasks as has been shown in the contemporary work by Rodriguez et al. [31] in the temporal moment localization task.

## 2   Related Work

**Vision-and-language navigation**   The Vision-and-language navigation problem has drawn significant research interest. Early work by Wang et al. [37] combines model-based and model-free reinforcement learning for navigation. The Speaker-Follower [9] enables self-supervised training by generating augmented samples and apply a panoramic action space for efficient navigation. Later, the Self-Monitoring agent [22] applies a visual-textual co-grounding network and a progress monitor to guide the transition of language attention, this work is further improved by path-scoring and backtracking methods, such as the Regretful agent [23] and the Tactical Rewind [18]. Wang et al. [36] propose a Self-Supervised Imitation Learning (SIL) method to pre-explore the environment refer to the speaker-generated instructions [9], which is later improved by the Environmental Dropout agent [32] trained with the A2C algorithm [24] and the proposed environmental dropout method. To enhance the alignment between the predicted and the ground-truth path, Jain et al. [17] and Ilharco et al. [15] design path similarity measurement and use it as rewards in reinforcement learning. Landi et al. [20] propose a fully-attentive agent that applies an early fusion to merge textual and visual clues and a late fusion for action prediction. Very recently, to enhance the learning of generic representations, AuxRN [40] apply various auxiliary losses to train the agent. PRESS [21] uses the large-scale pre-trained language model BERT [8] to better interpret the complex instructions and PREVALENT [10] pre-trains the vision and language encoders on a large amount of image-text-action triplets. Different from previous work which does not explicitly model the intra-modality relationships, we apply graph neural networks to exploit the semantic connection among the scene, its objects and the directional clues. As in the work by Hu et al. [13], our agent also grounds separately to different visual features, but their network is based on a mixture-of-expert fashion and applies test-time ensemble, which means, different experts are working independently as a voting system rather than cooperatively to reason the most probable future action.

**Graphs for relationship modelling**   Graph neural networks have been applied in a wide domain of problems for modelling the inter- and intra-modality relationships. Structural-RNN [16] constructs a spatial-temporal graph as a RNN mixture to model the relationship between human and object through time sequence to represent an activity. In vision-and-language research, graph representations are often applied on objects in the scene [33, 26, 35, 14]. Teney et al. [33] build graphs over the scene objects and over the question words, and exploit the structures in these representations for visual question answering. Hu et al. [14] propose a Language-Conditioned Graph Network (LCGN) where each node is initialized as contextualized representations of an object and it is updated through iterative message passing from the related objects conditioned on the textual input. Inspired by previous work, we build language and visual graphs to reason from sequential inputs. However, our graphs model the semantic relationships among distinct visual features (beyond objects) and are especially designed for navigation.

## 3   Language and Visual Entity Relationship Graph

In this section, we will discuss the proposed graph networks for interpreting navigational instruction and modelling the semantic relationships of distinct visual features. As summarized by Figure 1, at each navigational step, the agent first encodes the observed features and updates the state representation. Then, during decoding phase, the language attention graph performs a two-level soft-attention on the given instruction, where the first level grounds to specialized contexts and the second level grounds to relational contexts. Next, the language-conditioned visual graph will initialize the scene, the object and the directional features as nodes and enable message passing among the nodes for updating their contents. Finally, all the updated nodes are used for determine an action.

### 3.1   Problem definition and VLN backbone

In VLN on R2R, formally, given an instruction $\boldsymbol{w}$ as a sequence of $l$ words $\langle w_1, w_2, ..., w_l \rangle$, the agent needs to move on a connectivity graph which contains all the navigable points of the environment to reach the described target. At each time step $t$, the agent receives a panoramic visual input of its surrounding[2], which contains of 36 single view images as $\mathcal{V}^g \triangleq \langle \boldsymbol{v}_1^g, \boldsymbol{v}_2^g, ..., \boldsymbol{v}_{36}^g \rangle$. Within the

single views, there exists $n$ candidate directions where the agent can navigate to, we denote the visual features associated with those directions as $\langle \boldsymbol{v}_1^c, \boldsymbol{v}_2^c, ..., \boldsymbol{v}_n^c \mid \boldsymbol{v}_i^c \in \mathcal{V}^g \rangle$.

As in previous work [9, 32], we construct a 128-dimensional directional encoding $\boldsymbol{d}_i$ by replicating $(\cos\theta_i, \sin\theta_i, \cos\phi_i, \sin\phi_i)$ by 32 times to represent the orientation of each single view image with respect to the agent's current orientation, where $i$ refers to the index of an image, $\theta_i$ and $\phi_i$ are the angles of the heading and elevation. This directional encoding is then concatenated with the panoramic visual input to obtain the global visual features $\boldsymbol{f}_i^g = [\boldsymbol{v}_i^g; \boldsymbol{d}_i]$. For each candidate direction, we have $\langle \boldsymbol{d}_1^c, \boldsymbol{d}_2^c, ..., \boldsymbol{d}_n^c \rangle$, which will be used as directional features in the visual graph.

We also apply an object detector to extract $k$ objects for each candidate direction $i$, we denote the encoded object features as $\langle \boldsymbol{o}_{i,1}^c, \boldsymbol{o}_{i,2}^c, ..., \boldsymbol{o}_{i,k}^c \rangle$.

To obtain the language representations, the agent converts the given instruction $\boldsymbol{w}$ with a learned embedding as $\hat{\boldsymbol{w}}_j = \text{Embed}(w_j)$. Then, it applies a bidirectional-LSTM to encode the entire sentence as $\langle \boldsymbol{u}_1, \boldsymbol{u}_2, ..., \boldsymbol{u}_l \rangle = \text{Bi-LSTM}(\hat{\boldsymbol{w}}_1, \hat{\boldsymbol{w}}_2, ..., \hat{\boldsymbol{w}}_l)$, where $\boldsymbol{u}_j$ is the hidden state of word $w_j$ in the instruction.

To build a global awareness of the environment at each time step $t$, the agent grounds the previously attended global context $\tilde{\boldsymbol{h}}_{t-1}^g$ to the global visual features via soft-attention [38] to produce an attended global visual feature as $\tilde{\boldsymbol{f}}^g = \text{SoftAttn}(\tilde{\boldsymbol{h}}_{t-1}^g, \boldsymbol{f}^g)$. To keep track of the navigation process, we apply an instruction-aware LSTM to represent the agent current state as

$$\boldsymbol{h}_t = \text{LSTM}([\boldsymbol{a}_{t-1}; \tilde{\boldsymbol{f}}^g], \tilde{\boldsymbol{h}}_{t-1}^g) \tag{1}$$

where $\boldsymbol{a}_{t-1}$ is the directional encoding at the previously selected action direction, $\tilde{\boldsymbol{h}}_{t-1}^g$ is produced by the language attention graph, which will be discussed later.

## 3.2 Language Attention Graph

We propose a language attention graph where the nodes on the graph represents specialized contexts of the instruction and the edges model the relational context between specialized contexts.

**Nodes - specialized contexts** Consider an instruction such as "*With the sofa on your left, turn right into the bathroom and stop*", there are three essential and distinct type of information in the language which the agent needs to identify first: text relating to scene (bathroom), text relating to object/landmark (sofa) and text relating to direction (turn right into, stop). To this end, the network first performs three soft-attentions independently on the instruction, using the current agent state $\boldsymbol{h}_t$, to pick out the three specialized attended contexts as:

$$\tilde{\boldsymbol{h}}^s = \text{SoftAttn}^s(\boldsymbol{h}_t, \boldsymbol{u}) \qquad \tilde{\boldsymbol{h}}^o = \text{SoftAttn}^o(\boldsymbol{h}_t, \boldsymbol{u}) \qquad \tilde{\boldsymbol{h}}^d = \text{SoftAttn}^d(\boldsymbol{h}_t, \boldsymbol{u}) \tag{2}$$

where the superscripts indicate the corresponding type of text and different query projections, SoftAttn denotes the same soft-attention function [38].

We consider the specialized contexts $\tilde{\boldsymbol{h}}^s$, $\tilde{\boldsymbol{h}}^o$ and $\tilde{\boldsymbol{h}}^d$ as nodes in our language attention graph. Also, relate to the agent's state representation in Eq. (1), the attended global context $\tilde{\boldsymbol{h}}^g$ for the next time step is computed by averaging the current three specialized contexts.

**Edges - relational contexts** To interpret the instruction without ambiguity, it is important for the agent to understand the relationship between the specialized contexts. For instance, the direction "*turn right*" is conditioned on the scene "*bathroom*". To model such relationship, we construct edges between any two nodes in the graph by performing a second level language attention. Take the scene-related and direction-related texts as an example, we model the scene-direction relational context as

$$\tilde{\boldsymbol{h}}^{sd} = \text{SoftAttn}^{sd}(\boldsymbol{\Pi}_{sd}[\tilde{\boldsymbol{h}}^s; \tilde{\boldsymbol{h}}^d], \boldsymbol{u}) \tag{3}$$

where $\boldsymbol{\Pi}_{sd}$ and all $\boldsymbol{\Pi}$ in this section represent a learnable non-linear projection, with Tanh as the activation function. Similarly, we obtain the scene-object and object-direction context $\tilde{\boldsymbol{h}}^{so}$ and $\tilde{\boldsymbol{h}}^{od}$.

The language attention graph which contains three specialized contexts and three relational contexts is hence prepared for modelling the semantic relationships among visual features.

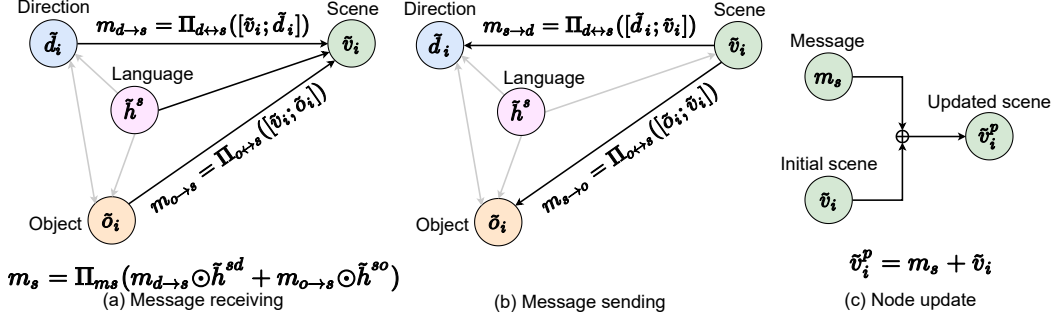

Figure 2: Message passing and update of the scene node.

## 3.3 Language-Conditioned Visual Graph

To perceive the rich visual clues in the environment referring to the instruction, we propose a language-conditioned visual graph as shown in Figure 1(b) to model the semantic relationship among different visual features and to reason the most probable action at each time step. Following the language graph, we defined the three visual features as scene, object and direction at each candidate direction $i$, corresponding to three observed features at that direction respectively: the image representation $\boldsymbol{v}_i^c$, the object feature $\boldsymbol{o}_i^c$ and the directional encoding of each image $\boldsymbol{d}_i^c$.

**Node Initialization**  We apply non-linear mappings to project different visual features to the same space and initialize them as the nodes in the visual graph. Specifically, to aggregate the object features of each candidate direction, we attend $\boldsymbol{o}_i^c$ with the specialized object context $\tilde{\boldsymbol{h}}^o$ from the language graph as $\tilde{\boldsymbol{o}}_i^c = \text{SoftAttn}(\tilde{\boldsymbol{h}}^o, \boldsymbol{o}_i^c)$. To enable a coherent flow of information over the graph at each navigational step, we define a temporal link of the graph over the direction node, as $\tilde{\boldsymbol{d}}_i^c = [\boldsymbol{d}_i^c; \tilde{\boldsymbol{d}}_{t-1}^p]$, where $\tilde{\boldsymbol{d}}_{t-1}^p$ is the updated direction node feature at the previously selected action direction. Meanwhile, we apply an element-wise product between each projected visual feature and the corresponding specialized context to build the first textual-visual connection of specialized contents of the two graphs:

$$\tilde{\boldsymbol{v}}_i = \boldsymbol{\Pi}_{sp}(\boldsymbol{v}_i^c)\odot\tilde{\boldsymbol{h}}^s \qquad \tilde{\boldsymbol{o}}_i = \boldsymbol{\Pi}_{op}(\tilde{\boldsymbol{o}}_i^c)\odot\tilde{\boldsymbol{h}}^o \qquad \tilde{\boldsymbol{d}}_i = \boldsymbol{\Pi}_{dp}(\tilde{\boldsymbol{d}}_i^c)\odot\tilde{\boldsymbol{h}}^d \qquad (4)$$

where $\odot$ is the element-wise product.

**Message passing**  To predict the correct action, the agent needs to understand the relationships among different visual clues according to the instruction. For instance, the agent should identify the scene once it enters the "*bathroom*" and perform the action "*stop*". To this end, we model the relationship among the scene, the object and the direction features through language-conditioned message passing, where each node receives and sends information from and to the other nodes through the edges on the visual graph. To be concise and clear, we explain the idea using the scene node. As shown in Figure 2(a), the scene node receives messages $\boldsymbol{m}_{d\to s}$ and $\boldsymbol{m}_{o\to s}$ from the direction node and object node respectively, where

$$\boldsymbol{m}_{d\to s} = \boldsymbol{\Pi}_{d\leftrightarrow s}([\tilde{\boldsymbol{v}}_i; \tilde{\boldsymbol{d}}_i]) \qquad \boldsymbol{m}_{o\to s} = \boldsymbol{\Pi}_{o\leftrightarrow s}([\tilde{\boldsymbol{v}}_i; \tilde{\boldsymbol{o}}_i]) \qquad (5)$$

To guide the messages for delivering relevant information to the scene node, we condition each message on the corresponding relation context produced by the language graph. Similar as previous, we apply an element-wise product between each message and the relation context to build the second textual-visual connection of relation contents of the two graphs as:

$$\boldsymbol{m}_s = \boldsymbol{\Pi}_{ms}(\boldsymbol{m}_{d\to s}\odot\tilde{\boldsymbol{h}}^{sd} + \boldsymbol{m}_{o\to s}\odot\tilde{\boldsymbol{h}}^{so}) \qquad (6)$$

Meanwhile, each node shares information to the other nodes. As shown in Figure 2(b), the direction node and the object node receive messages $\boldsymbol{m}_{s\to d}$ and $\boldsymbol{m}_{s\to o}$ from the scene node respectively as:

$$\boldsymbol{m}_{s\to d} = \boldsymbol{\Pi}_{d\leftrightarrow s}([\tilde{\boldsymbol{d}}_i; \tilde{\boldsymbol{v}}_i])\odot\tilde{\boldsymbol{h}}^{sd} \qquad \boldsymbol{m}_{s\to o} = \boldsymbol{\Pi}_{o\leftrightarrow s}([\tilde{\boldsymbol{o}}_i; \tilde{\boldsymbol{v}}_i])\odot\tilde{\boldsymbol{h}}^{so} \qquad (7)$$

Notice that, the message passing weights $\boldsymbol{\Pi}_{d\leftrightarrow s}$ and $\boldsymbol{\Pi}_{o\leftrightarrow s}$ are shared between the sending and the receiving paths, $\boldsymbol{\Pi}_{o\leftrightarrow d}$ likewise.

**Node update**   Each node synchronously receives relevant information from the other nodes to enrich its content. As shown in Figure 2(c), each node is updated by summing the received message and its initial node feature as:

$$\tilde{\boldsymbol{v}}_i^p = \boldsymbol{m}_s + \tilde{\boldsymbol{v}}_i \qquad\qquad \tilde{\boldsymbol{o}}_i^p = \boldsymbol{m}_o + \tilde{\boldsymbol{o}}_i \qquad\qquad \tilde{\boldsymbol{d}}_i^p = \boldsymbol{m}_d + \tilde{\boldsymbol{d}}_i \qquad\qquad (8)$$

**Action prediction**   Finally, we infer the action probability for each candidate direction $i$ from the three updated nodes by

$$p_i = \text{Softmax}_i(\boldsymbol{W}_{logit}([\tilde{\boldsymbol{v}}_i^p; \tilde{\boldsymbol{o}}_i^p; \tilde{\boldsymbol{d}}_i^p])) \qquad\qquad (9)$$

where $\boldsymbol{W}_{logit}$ is a learnable linear mapping.

### 3.4   Training

We apply the *Imitation Learning (IL) + Reinforcement Learning (RL)* objectives [32] to train our network. In imitation learning, the agent takes the teacher action $a_t^*$ at each time step to efficiently learn to follow the ground-truth trajectory. In reinforcement learning, the agent samples an action $a_t^s$ from the action probability $p_t$ and learns from the rewards, which allows the agent to explore the environment and improve generalizability. Combining IL and RL balances exploitation and exploration when learning to navigate, formally, we have:

$$\mathcal{L} = \lambda \underbrace{\sum_{t=1}^{T} -a_t^* \log(p_t)}_{\mathcal{L}_{IL}} + \underbrace{\sum_{t=1}^{T} -a_t^s \log(p_t) A_t}_{\mathcal{L}_{RL}} \qquad\qquad (10)$$

where $\lambda$ is a coefficient for weighting the IL loss, $T$ is the total number of step the agent chooses to take and $A_t$ is the advantage in A2C algorithm [24].

## 4   Experiments

### 4.1   Setup

**Datasets**   The Room-to-Room (R2R) dataset [3] consists of 10,567 panoramic views in 90 real-world environments as well as 7,189 trajectories where each is described by three natural language instructions. Viewpoints in an environment are defined over a connectivity graph, the agent can navigate in the environment using the Matterport3D Simulator [4]. The dataset is split into train, validation seen, validation unseen and test unseen sets. To show the generalizability of our proposed agent, we also evaluate the agent's performance on the Room-for-Room (R4R) dataset [17], an extended version of R2R with longer instructions and trajectories. The VLN task on R2R and R4R is to test the agent's performance in novel environments with new instructions.

**Evaluation metrics**   We follow the standard metrics employed by the previous work in R2R to evaluate the performance of our proposed agent. These include the Trajectory Length (TL) of the agent's navigated path in meters, the Navigation Error (NE) which is the average distance in meters between the agent's final position and the target, the Success Rate (SR) which is ratio of the agent stopping at 3 meters within the target, and the Success Rate weighted by the normalized inverse of the Path Length (SPL) [1]. For the R4R data [17], we employ three extra metrics to measure the path fidelity (similarity between the ground-truth path and the predicted path), including the Coverage weighted by Length Score (CLS) [17], the normalized Dynamic Time Warping (nDTW) [15] and the Success weighted by nDTW (SDTW) [15]. The most important metric of the R2R results is SPL, since it measures both the accuracy and the efficiency of navigation. As for the R4R, path fidelity measurements are important metrics for evaluating how much an agent understands and follows the instructions.

**Implementation details**   We apply the image representations encoded by pre-trained ResNet-152 [11] on Places365 [39] provided in R2R [3] as the scene features. We also apply a Faster-RCNN [30] pre-trained on the Visual Genome Dataset [19] to extract 36 object labels for each candidate directions, encode them using GloVe [27] and use them as the object features. Notice that the object detector trained by Anderson et al. [2] classifies an object among 1,600 categories.

Table 1: Comparison of single-run performance with the state-of-the-art methods on R2R. †: work that applies pre-trained textual or visual encoders.

| | Validation Seen | | | | Validation Unseen | | | | Test Unseen | | | |
|---|---|---|---|---|---|---|---|---|---|---|---|---|
| Agent | TL | NE↓ | SR↑ | SPL↑ | TL | NE↓ | SR↑ | SPL↑ | TL | NE↓ | SR↑ | SPL↑ |
| Random | 9.58 | 9.45 | 0.16 | - | 9.77 | 9.23 | 0.16 | - | 9.89 | 9.79 | 0.13 | 0.12 |
| Human | - | - | - | - | - | - | - | - | 11.85 | 1.61 | 0.86 | 0.76 |
| Seq2Seq [3] | 11.33 | 6.01 | 0.39 | - | 8.39 | 7.81 | 0.22 | - | 8.13 | 7.85 | 0.20 | 0.18 |
| Speaker-Follower [9] | - | 3.36 | 0.66 | - | - | 6.62 | 0.35 | - | 14.82 | 6.62 | 0.35 | 0.28 |
| SMNA [22] | - | **3.22** | 0.67 | 0.58 | - | 5.52 | 0.45 | 0.32 | 18.04 | 5.67 | 0.48 | 0.35 |
| RCM+SIL (train) [36] | 10.65 | 3.53 | 0.67 | - | 11.46 | 6.09 | 0.43 | - | 11.97 | 6.12 | 0.43 | 0.38 |
| Are-You-Looking [13] | - | - | - | - | - | - | 0.52 | - | - | - | - | - |
| Regretful [23] | - | 3.23 | 0.69 | 0.63 | - | 5.32 | 0.50 | 0.41 | 13.69 | 5.69 | 0.48 | 0.40 |
| FAST (short) [18] | - | - | - | - | 21.17 | 4.97 | 0.56 | 0.43 | 22.08 | 5.14 | 0.54 | 0.41 |
| PRESS [21] † | 10.57 | 4.39 | 0.58 | 0.55 | 10.36 | 5.28 | 0.49 | 0.45 | 10.77 | 5.49 | 0.49 | 0.45 |
| EnvDrop [32] | 11.00 | 3.99 | 0.62 | 0.59 | 10.70 | 5.22 | 0.52 | 0.48 | 11.66 | 5.23 | 0.51 | 0.47 |
| AuxRN [40] | - | 3.33 | **0.70** | **0.67** | - | 5.28 | 0.55 | 0.50 | - | 5.15 | **0.55** | 0.51 |
| PREVALENT [10] † | 10.32 | 3.67 | 0.69 | 0.65 | 10.19 | **4.71** | **0.58** | **0.53** | 10.51 | 5.30 | 0.54 | 0.51 |
| Ours | 10.13 | 3.47 | 0.67 | 0.65 | 9.99 | 4.73 | 0.57 | **0.53** | 10.29 | **4.75** | **0.55** | **0.52** |

Table 2: Comparison of single-run performance with the state-of-the-art methods on R4R. *goal* indicates distance reward and *fidelity* indicates path similarity reward in reinforcement learning.

| | Validation Seen | | | | | | Validation Unseen | | | | | |
|---|---|---|---|---|---|---|---|---|---|---|---|---|
| Agent | NE↓ | SR↑ | SPL↑ | CLS↑ | nDTW↑ | SDTW↑ | NE↓ | SR↑ | SPL↑ | CLS↑ | nDTW↑ | SDTW↑ |
| Speaker-Follower [9] | 5.53 | 0.52 | 0.37 | 0.46 | - | - | 8.47 | 0.24 | 0.12 | 0.30 | - | - |
| RCM-a (goal) [17] | 5.11 | 0.56 | 0.32 | 0.40 | - | - | 8.45 | 0.29 | 0.10 | 0.20 | - | - |
| RCM-a (fidelity) [17] | 5.37 | 0.53 | 0.31 | 0.55 | - | - | 8.08 | 0.26 | 0.08 | 0.35 | - | - |
| RCM-b (goal) [15] | - | - | - | - | - | - | - | 0.29 | 0.15 | 0.33 | 0.27 | 0.11 |
| RCM-b (fidelity) [15] | - | - | - | - | - | - | - | 0.29 | 0.21 | 0.35 | 0.30 | 0.13 |
| PTA (high-level) [20] | **4.54** | **0.58** | 0.39 | **0.60** | 0.58 | 0.41 | 8.25 | 0.24 | 0.10 | 0.37 | 0.32 | 0.10 |
| Ours | 5.31 | 0.52 | **0.46** | 0.55 | **0.62** | **0.50** | 7.43 | **0.36** | **0.26** | **0.41** | **0.47** | **0.34** |

To simplify the object vocabulary and remove rare detections, we combine the 1,600 classes to 101 classes, where the 100 classes are the most frequent objects appear in both the instruction and the environment of the training data, and the remaining 1 class is *others*.

The training of our agent consists of two stages. At the first stage, we train the agent on the train split of the dataset. At the second stage, we pick the model with the highest SPL from the first stage and keep training it with self-supervised learning [32] on the augmented data generated from the train set [32]. During self-supervised learning, we apply the same Speaker module with environmental dropout proposed in the baseline method [32]. As for the R4R experiment, only the first stage of training is performed for a fair comparison with the previous methods. Moreover, although our graph network support multi-iteration updates, we found in experiments that more than one iteration results in similar performance, hence for efficiency, we only apply a single-step update. We refer the Appendix for more visual features processing and model training details.

## 4.2 Results and Analysis

**Comparison with SoTA**   Agent's performance in single-run (greedy search) setting reflects its efficiency in navigation as well as its generalizability to novel instructions and environments. As shown in Table 1, on the R2R benchmark, our method significantly outperforms the baseline method EnvDrop [32], obtaining 5% absolute improvement on SPL on the two unseen splits. On the validation unseen split, our method achieves very similar performance as the concurrent work PREVALENT [10], which exceeds previous methods on SR and SPL by a large margin. On the test split, our method achieves the best result on NE, SR and SPL. The navigation error is 0.39m shorter than the previous best, indicating that on average our agent navigates much closer to the target. On the R4R dataset (Table 2), our method significantly outperforms the previous state-of-the-arts over all the metrics on the validation unseen split. The nDTW and SDTW are absolutely increased by 15% and 21%, indicating that our agent follows the instruction better and navigates on the described path to reach the target.

Table 3: Ablation study showing the effect of different visual clues and the relationship modelling in the graph networks. *Graph* with a checkmark indicates edges exist among nodes.

| Model | Scene | Object | Direction | Language | Visual | TL | NE↓ | SR↑ | SPL↑ | TL | NE↓ | SR↑ | SPL↑ |
|---|---|---|---|---|---|---|---|---|---|---|---|---|---|
| | \multicolumn Clues | | | Graph | | Validation Seen | | | | Validation Unseen | | | |
| Baseline [32] | | | | | | 11.00 | 3.99 | 0.62 | 0.59 | 10.70 | 5.22 | 0.52 | 0.48 |
| 1 | ✓ | | ✓ | | | 10.16 | 3.60 | 0.64 | 0.61 | 10.65 | 4.84 | 0.54 | 0.51 |
| 2 | ✓ | | ✓ | ✓ | ✓ | 12.66 | 3.57 | 0.66 | 0.63 | 17.05 | 4.82 | 0.55 | 0.50 |
| 3 | | ✓ | ✓ | ✓ | ✓ | 16.34 | 5.23 | 0.62 | 0.48 | 17.94 | 5.23 | 0.51 | 0.46 |
| 4 | ✓ | ✓ | ✓ | | | 9.86 | 3.93 | 0.62 | 0.59 | 9.42 | 4.79 | 0.54 | 0.51 |
| 5 | ✓ | ✓ | ✓ | | ✓ | 12.77 | 3.85 | 0.64 | 0.61 | 17.10 | 4.77 | 0.55 | 0.50 |
| Full model | ✓ | ✓ | ✓ | ✓ | ✓ | 10.13 | **3.47** | **0.67** | **0.65** | 9.99 | **4.73** | **0.57** | **0.53** |

Table 4: Comparison on success rate of models with and without object visual clues. *Only the groups of instructions with more than 30 samples are shown.

| Model | Number of Objects in Instruction* | | | | | | | | | |
| | 1 | 2 | 3 | 4 | 5 | 6 | 7 | 8 | 9 | Avg |
|---|---|---|---|---|---|---|---|---|---|---|
| w/o object clues (Model #2) | **0.504** | 0.566 | 0.589 | 0.565 | 0.501 | 0.527 | 0.552 | 0.484 | 0.405 | 0.528 |
| with object clues (Full model) | **0.504** | 0.586 | 0.591 | 0.578 | 0.530 | 0.575 | 0.584 | 0.531 | 0.459 | 0.569 |
| Difference (+) | 0.000 | 0.020 | 0.002 | 0.023 | 0.029 | 0.048 | 0.032 | 0.047 | 0.054 | 0.041 |

**Ablation experiment**   Table 3 presents the contribution of different components in the graph networks to the navigation results on R2R. Comparing Model #1 to the baseline model, we can see that specializing the textual and visual clues boosts the agent's performance. Results of Model #1, #4 and the full model indicate that the object features do not benefit the navigation unless the relationship modelling is introduced. Comparison of Model #3 and the full model shows that the scene features are essential, since it contains the important information of the agent's surroundings. Model #4 and Model #5 shows the importance of the textual-visual connections between the two graphs, without such connections, the agent can only achieve slight improvement over the agent without graphs. Finally, comparing Model #2 and the full model, the trajectory length is significantly decreased by introducing object features, it is likely that this is resulting from the strong localization signals provided by the objects along the path.

We further investigate the unseen samples solved by our model with object visual clues (Full model) and without object visual clues (Model #2) in Table 4. First, for each instruction in validation unseen split, we extract the a list of *nouns* using the Stanford NLP Parser [28] and filter it with the object vocabulary with 101 categories. We consider the resulting list of *nouns* as the number of objects in an instruction. Then, for each group of instructions, we evaluate the success rate of Model #2 and the Full model, and compare their difference. As shown in the table, the Full model performs better than Model #2 in almost all the instruction groups. And in general, as the number of objects increases, the increment of success rate becomes larger for the Full model. This result indicates that our model with object visual clues can utilize the object textual clues better, and it is able to solve instructions with more objects with a higher success rate.

Finally, we observe in our experiments that there exists a large difference between the subset of unseen samples solved by our graph networks with objects and the subset solved by our graph networks without objects; about 10% of samples can be uniquely solved by the Full model, whereas another 8% of samples can be uniquely solved by model #2. It will be an interesting future work to allow the agent to combine and take advantage of the two models to solve more novel samples.

**Visualization of language attention**   Figure 3 presents the distribution of attention weights of the specialized and the relational context at the first navigational step. We can see that all the weights are mainly distributed at the early part of the sentence, which agrees with the navigation progress. In terms of the specialized context, attentions on scene and object mainly cover the text around the words "*stairs*" and "*exit sign*" respectively, while the attention on direction focuses only on the action-related term "*walk down*". As for the relational context, attention on scene-object considers the specialized contexts and decides that "*stairs*" should be a more important clue. Although the "*exit sign*" is not yet in the vision, the object-direction context is suggesting that the agent should look for an "*exit sign*" as it walks down the stairs. We refer the Appendix for more examples.

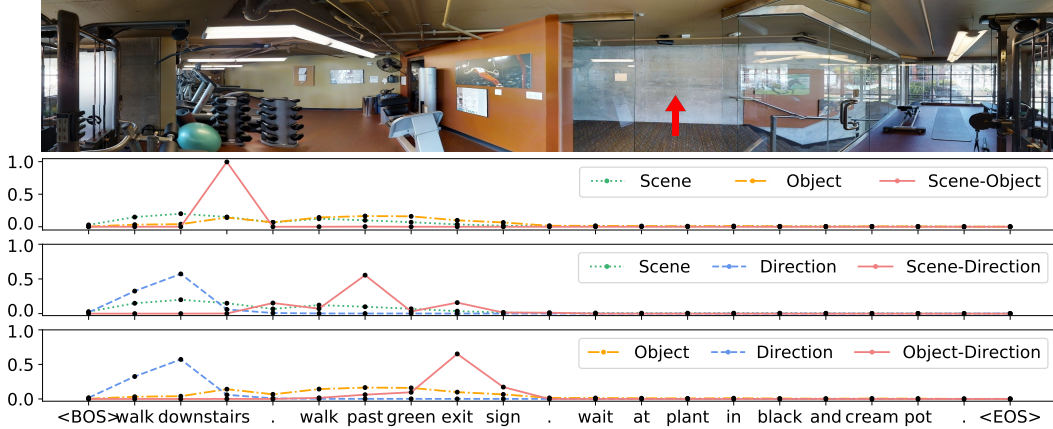

Figure 3: Distribution of language attention weights of the specialized and the relational contexts at the first navigational step. Red arrow in the image is the predicted direction.

# 5  Conclusion and Future Direction

In this paper, we present a novel language and visual entity relationship graph to exploit the connection among the scene, its objects and directional clues during navigation. Learning the relationships helps clarifying ambiguity in the instruction and building a comprehensive perception of the environment. Our proposed graph networks for VLN improves over the existing methods on the R2R and R4R benchmark, and becomes the new state-of-the-art.

**Future direction**  Objects mentioned in the instruction are important landmarks which can benefit the navigation by: allowing the agent to be aware of the exact progress of completing the instruction, providing strong localization signals to the agent in the environment and clarifying ambiguity for choosing a direction. However, in this work, we only apply objects as another visual features, rather than use them for progress monitoring, instance tracking or reward shaping in reinforcement learning. We believe that there is a great potential of using objects and graph networks for relationship modelling in future research on vision-and-language navigation.

## Broader Impact

The R2R [3] and R4R [17] datasets applied in our research contain a large number of photos of indoor environments, freely available under license from Matterport3D. None of the photos contain recognizable individuals. All experiments are performed on the Matterport3D Simulator [4], which are safe and confidential. This research is at the early stages of pushing towards robots that follows human instructions. In the future, if a such robot is implemented in real-world, it could benefit the society by assisting people to finish daily work. There are minimal ethical, privacy or safety concerns.

## Funding and Competing Interests

This work was partially funded by the Australian Research Council Centre of Excellence in Computer Vision (CE140100016).

## Footnotes

[1]VLN Leaderboard: https://evalai.cloudcv.org/web/challenges/challenge-page/97/overview

[2]In the following sections, for brevity of notation we drop the time index $t$ except when updating the agent's state between two time steps; when omitted all entities are assumed to be in the same time step.

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
