[Supplementary Material]

# Appendix: Language and Visual Entity Relationship Graph for Agent Navigation

**Yicong Hong**[1]    **Cristian Rodriguez-Opazo**[1]    **Yuankai Qi**[2]    **Qi Wu**[2]    **Stephen Gould**[1]

[1]Australian National University    [2]The University of Adelaide
[1,2]Australian Centre for Robotic Vision

{yicong.hong, cristian.rodriguez, stephen.gould}@anu.edu.au
qykshr@gmail.com, qi.wu01@adelaide.edu.au

## Appendix A    Equations

We provide the equations of the soft-attention function SoftAttn() and the non-linear projection $\mathbf{\Pi}()$ which are repeatedly used in this paper.

### A.1    Soft-Attention

Suppose $\boldsymbol{q}$ is the query and $\boldsymbol{y} = \{\boldsymbol{y}_1, \boldsymbol{y}_2, ..., \boldsymbol{y}_n\}$ is a set of values, we express the soft-attention [11] function $\tilde{\boldsymbol{y}} = \text{SoftAttn}(\boldsymbol{q}, \boldsymbol{y})$ as

$$\alpha_j = \text{softmax}_j(\boldsymbol{y}_j^T \boldsymbol{W} \boldsymbol{q}) \tag{1}$$

$$\tilde{\boldsymbol{y}} = \sum_{j=1}^{n} \alpha_j \boldsymbol{y}_j \tag{2}$$

where $\boldsymbol{W}$ is a $\mathbb{R}^{m \times n}$ learned linear projection for $n$-dimensional query and $m$-dimensional values, $\alpha_j$ is the attention weight of value $\boldsymbol{y}_j$ and $\tilde{\boldsymbol{y}}$ is the attended value.

### A.2    Non-Linear Projection

Suppose $\boldsymbol{y}$ is an input vector, the non-linear projection $\mathbf{\Pi}(\boldsymbol{y})$ is defined as:

$$\mathbf{\Pi}(\boldsymbol{y}) = \text{Tanh}(\boldsymbol{W} \boldsymbol{y}) \tag{3}$$

where $\boldsymbol{W}$ is a learned linear projection, and Tanh is applied element-wise.

## Appendix B    Implementation Details

### B.1    Observed Features

At each navigable viewpoint, the agent observes a panoramic view of its surroundings. The panoramic view consists of 36 single-view images at 12 heading angles and 3 elevation angles relative to agent's orientation. The scene, object and directional features are encoded for each single-view separately.

**Scene features**    The R2R dataset [2] provides image features for each single-view encoding by two convolutional networks, a ResNet-152 [4] pre-trained on ImageNet [9] and a ResNet-152 [4] pre-trained on Places365 [12]. Ma et al. finds that the Self-Monitoring agent [6] performs similar with the two image features, we also find in our experiments with the baseline model (EnvDrop [10]) that using the two image features achieve very similar results. However, while most of the previous

works apply ImageNet features as the scene feature, our agent uses the Places365 features for two reasons: (1) the Places365 dataset is proposed mainly for scene recognition task, which closely agrees with the concept of scene that we defined in this paper. (2) The ImageNet features can be considered as a summary of salient objects in the scene, which contains similar information as the object representations applied by our agent.

**Object features**   In this work, we apply a Faster-RCNN [8] pre-trained on the Visual Genome [5] by Anderson et al. [1] as object detector. We use the detector to predict object labels rather than to extract object visual encodings (convolutional features). Then, we encode the labels using GloVe [7] and use the encoded labels as object features. We argue that using labels instead of visual encodings can provide much certain signals to indicate the existence of an object, since the agent only needs to learn the text-text correspondence rather than the hard text-visual correspondence. We experiment with the encoded object labels and the object visual encodings, and find that the agent performs much better with the encoded object labels.

**Directional features**   As in previous work [3, 6, 10], we apply a 128-dimensional directional encoding by replicating $(\cos\theta_i, \sin\theta_i, \cos\phi_i, \sin\phi_i)$ by 32 times to represent the orientation of each single-view $i$ with respect to the agent's current orientation, where $\theta_i$ and $\phi_i$ are the angles of the heading and elevation to that single-view. Replicating the encoding by 32 times does not enrich its information but makes its gradient 32 times larger during back-propagation. We suspect that this benefits the agent to learn about the action-related terms (e.g. "*turn left*, "*go forward*") in the instructions, which are sometimes more important than the other visual clues.

## B.2   Training

The training of our agent consists of two stages. At the first stage, we train the agent on the train split of the dataset. At the second stage, we pick the model with the highest SPL from the first stage and train with self-supervised learning on the augmented data generated from the train set [10]. In the second stage, we also perform a one-step learning rate decay. Once the agent's performance saturates, we pick the model with the highest SPL and continue training with learning rate of 1e-5.

**Hyper-parameters**   Most of the hyper-parameters and settings in our network are the same as in the baseline model [10], such as the learning rate (1e-4 before decay), the maximum navigational step allowed (35 steps), the weighting of imitation learning loss (0.2 on training data and 0.6 on augmented data) and the optimizer (RMSprop). We reduce the batch size from 64 to 32 due to the GPU memory limitation. We did not perform hyper-parameter search on the number of objects in each single-view or the dimension of features.

**Training iterations and evaluation**   We perform the first training stage for 80,000 iterations and continue the training in the second stage until 200,000 iterations. Then, after learning rate decay, we keep training the network until 300,000 iterations. Evaluation on the validation splits is performed every 100 iterations, results and network weights are carefully saved. We choose model with the highest SPL in the validation unseen split as the final model.

**Computing infrastructure and runtime**   All experiments are conducted on a NVIDIA RTX 2080 Ti GPU. The average runtime of the first training stage is 24 hours, and the second training stage requires at most 60 hours to complete (including the time spent on evaluation). Early stop has been applied if the network overfits and the agent's performance degenerates significantly.

## Appendix C   Graph with Muti-Iterations Update

Our proposed language-conditioned visual graph supports update with multiple iterations. In Table 1, we compare the results by training the agent with $N$ iterations of graph update and testing with $n \leq N$ iterations of graph update. Overall, as the number of iterations in training increases, agent's performance in testing drops. The agent trained and tested with a single-step graph update obtains the highest SR and SPL, hence for efficiency and better performance, the visual graph only run a single-step update.

Table 1: Comparison of single-run performance on R2R validation unseen split with different number of graph iteration in training and testing.

| | Iterations in Testing | | | | | | | | | | | |
| | 1 | | | | 2 | | | | 3 | | | |
| Iterations in Training | TL | NE↓ | SR↑ | SPL↑ | TL | NE↓ | SR↑ | SPL↑ | TL | NE↓ | SR↑ | SPL↑ |
|---|---|---|---|---|---|---|---|---|---|---|---|---|
| 1 | 9.99 | 4.73 | **0.57** | **0.53** | - | - | - | - | - | - | - | - |
| 2 | 9.84 | **4.60** | 0.56 | 0.52 | 15.04 | 4.62 | 0.56 | 0.52 | - | - | - | - |
| 3 | 9.71 | 4.79 | 0.54 | 0.50 | 10.17 | 4.90 | 0.54 | 0.51 | 13.06 | 4.78 | 0.55 | 0.51 |

## Appendix D  Visualization of Trajectories

In the following pages, we present some navigational trajectories and the attended relational contexts of our agent in the R2R validation unseen environments. In the figures, *Distance* is the current distance to target, red arrow indicates the predicted directions and red square indicates the predicted *STOP* action.