[Reviews · NeurIPS 2020]

Review 1

Summary and Contributions: The authors proposed a Language and Visual Entity Relationship Graph model for VLN. This model comes with the usage of the scene, object, and directional features, in which object features (extracted by Faster-RCNN pre-trained on Visual Genome) were not used in previous works. The authors show that adding these object features won't naive bring benefit for the VLN agent, rather it needs to be combined with the other features through their designed Relational Graph model. The authors show that on both R2R and R4R datasets they outperformed existing methods. Note that, in my opinion, their performance is slightly better on R2R if only compared to AuxRN [36] on SR and SPL on the Val unseen and test set, but much better on the NE metric.

Strengths: - A comprehensive evaluation of both R2R and R4R datasets, which use CLS, nDTW, and SDTW evaluation metrics. - Nice ablation study on the design choices of the model - The proposed method seems to use the Faster-RCNN pre-trained from Visual Genome and GloVe. One would expect that using additional object features is no doubt going to improve the navigation performance, but the ablation on Model #1 and #4 shows this can only improve navigation performance if relationship modeling is introduced. - Future direction subsection at the end of paper is nice and indicate some of the ideas worth to be explored for the research community.

Weaknesses: - Model seems to be a bit complicated (but this complicated design comes with ablation study)

Correctness: Yes, I believe the claims on how the authors proposed a new model that leverages the graph structure to use scene, object, and directional features is valid. This is verified by the ablation study and the experimental results on R2R and R4R datasets.

Clarity: Yes, overall the paper is well-written. I find it not as easy to follow but also not too difficult to figure out. Some of the things need to be further clarified as I mentioned in the other comments, but overall it's clear enough to understand the core concept of the proposed method.

Relation to Prior Work: Yes, the paper clearly indicates how their work/contributions are different from previous works.

Reproducibility: Yes

Additional Feedback: Yes, the authors provided implementation details in the Appendix and others should be able to replicate their reported results.


Review 2

Summary and Contributions: This paper presents a graph network-based visual and textual co-attention mechanism for Vision-and-Language navigation (where a virtual agent carries out natural language navigational instructions in a simulated photorealistic environment). The graph network has nodes for environment features taken from a pre-trained object detector, ResNet features from an object classification model, and directional features (i.e. agent heading); and corresponding nodes for text representations produced using attention over the instruction. Message passing is used to aggregate this information between nodes to produce the agent's action at each timestep. The method obtains very strong results on the R2R and R4R benchmark datasets, performing as well or better than the latest state-of-the-art models. --- update after the author response --- Thanks to the authors for the response! After it, and some discussion, I feel more positively and have raised my score to a 7 (from a 6). I'm optimistic that other work would be likely to build on this model, since 1) the model's lightweight in comparison to PREVALENT, but does as well or better 2) the graph structure of the model (over types of context representation) seems like it could make it easy to extend to and analyze other types of context representation. I also feel that the implementation and the design of the model -- getting it to work well in a seemingly large space of possible options -- are valuable contributions.

Strengths: I found the method well-motivated. Past work has explored using object detections for VLN, with mixed success; this paper shows how they can be used more effectively within an integrated model. The architecture will likely be of interest to researchers working on other related datasets and tasks (e.g. vision-and-dialog navigation, Touchdown VLN) and other methods which this architectural improvement could potentially benefit (e.g. navigation graph search methods, and continuous navigation). The paper does thorough ablation experiments, which motivate each of the design choices and give some indication of why past work using object representations has had mixed results.

Weaknesses: The paper's contribution is very focused, presenting an application of graph networks to visual features which have been used in past work, as the paper acknowledges. Because of this, I felt the evaluation could be broadened, as this paper's primary contribution is a better method for learning the language--vision relationship -- and both datasets evaluated on, R2R and R4R, have the same underlying visual representations and instructions (R4R just has longer paths and concatenated instructions). It would strengthen the paper to show results on another VLN dataset (e.g. Touchdown, Chen et al.), or another language navigation task (e.g. vision-and-dialog navigation, Thomason et al. 2019).

Correctness: I didn't see any issues with correctness. The evaluation follows the standard for these benchmarks.

Clarity: I found the paper overall clear in its description of the method. Equation (1) was a bit confusing: is h^g_{t-1} fed as an input to the LSTM (e.g. concatenated with a_{t-1} and f^g), or does it replace the recurrent hidden state typically used in the LSTM? It seems to be an attended representation of the sentence (from line 140), rather than a recurrent state, but the notation here was confusing. The ablation description would also be clearer if it named the specific terms which are removed in the graph ablations: what terms does action prediction (Eq 9) condition on when the language graph and visual graph are removed?

Relation to Prior Work: I found the description and comparison to prior work on VLN clear. One additional work using object-based representations for VLN is Zhang et al. 2020, Diagnosing the Environment Bias in Vision-and-Language Navigation.

Reproducibility: Yes

Additional Feedback: It would help to clarify earlier in the paper that the graph used is over visual observations from a single timestep in the trajectory (with the exception of the recurrent direction features), as opposed to passing messages over the navigation graph itself. If I understand Eq (1) correctly, the object features are not being fed as inputs into the recurrent LSTM decoder (just the scene and direction features) -- is this correct? If so, did the paper also explore feeding the attended object features? 198: "Combining IL and RL balances exploitation and exploration when learning to navigate": this was unclear to me. I'm interpreting it as meaning that IL somehow encourages more exploitation, but I'm unsure. Do any of the results in the ablation study ablate the temporal link on the direction node (mentioned in line 164)? In line 280, "there exists a large difference between the subset": I didn't see these results present in the paper; if they are it'd be worth having a more explicit reference to them here. The paper could benefit from proofreading for grammar.


Review 3

Summary and Contributions: The authors propose a novel Language and Visual Entity Relationship Graph for modelling the inter-modal relationships between text and vision, and the intra-modal relationships among visual entities.

Strengths: The authors propose a novel language and visual entity relationship graph for vision-and-language navigation that explicitly models the inter- and intra-modality relationships among the scene, the object, and the directional clues.

Weaknesses: 1. The rank in VLN Leaderboard is 22-th. The results of this proposed method is not better than existing methods, e.g., [1,2,3]. [1] Zhu F, Zhu Y, Chang X, et al. Vision-language navigation with self-supervised auxiliary reasoning tasks[C]//Proceedings of the IEEE/CVF Conference on Computer Vision and Pattern Recognition. 2020: 10012-10022. [2] Tan, Hao, Licheng Yu, and Mohit Bansal. "Learning to navigate unseen environments: Back translation with environmental dropout.". [3] Ke, Liyiming, et al. "Tactical rewind: Self-correction via backtracking in vision-and-language navigation." Proceedings of the IEEE Conference on Computer Vision and Pattern Recognition. 2019. 2. Are the parameter in Eq.(10) the same for different tasks? 3. The novelty and contribution of this work need to be further clarified. 4. The theoretical depth of the paper is limited. With respect to why the proposed framework outperforms the other models, more profound theoretical analysis and demonstration are required to be provided. 5. The computation complexity of the proposed algorithm should be analyzed and compared with other recently published state-of-the-art algorithms.

Correctness: Good.

Clarity: Good.

Relation to Prior Work: Good.

Reproducibility: Yes

Additional Feedback:


Review 4

Summary and Contributions: This paper proposes a novel vision-and-language navigation (VLN) model that learns the language and visual entity relational graph. It utilizes multiple features including scene, object, and directional features. The proposed model achieves new SOTA results on R2R and R4R datasets.

Strengths: - This paper tackles the VLN task by constructing a text-visual graph, which is novel in the VLN space though not new in the general vision-and-language research area. - This paper utilizes multiple effective features including scene, object and directional features and learns their relationship conditioned on the language input. - The model achieves SOTA results on R2R and R4R datasets. - In general, the paper is well written and easy to follow.

Weaknesses: - The proposed method is tailored for VLN and may limit its generalization to other domains (it is not new for other vision-and-language tasks). - Equation 2 in Section 3.2 is very confusing. If the same h_t and u are feed into the three attentions, how could different contexts be learned? It seems impossible. There seems to be something wrong, either the technique or the notations. - It seems only results of single runs are reported. However, VLN models may be sensitive to hyper-parameter tuning. It would be better if the authors can demonstrate the mean and standard deviation of multiple runs. - Error analysis is missing in the paper. In what cases the proposed model would fail? Can the authors provide some error analysis? - How would the model perform if it is allowed to pre-explore the unseen environments? It seems building a graph would also facilitate the effectiveness of pre-exploration, which is indeed an important setting in many cases. - In Table 3, it is unclear to me how Model #4 and #5 shows the importance of the connections between two graphs. - In Table 3, Model 3-6 also use the object features, but Model 4 and 6 have an efficient average path while the others don't. Why? - What is the self-supervised learning method mentioned in Line 231 as it is not introduced elsewhere? -------------------------------- after rebuttal ---------------------------------- After reading the authors' response and the other reviews, my major concerns have been addressed. The fact that this work is tailored for VLN is rather a minor comment than a major concern to me, as I believe that VLN is an interesting and important research area that deserves more attention and further study. But I think the authors could provide more insights of improving VLN or the general vision-language grounding with more in-depth discussions on the experimental results. I am inclined to accept this paper and would like to raise my score to 7. I hope the authors can include error analysis and average of multiple runs in the future version.

Correctness: Yes.

Clarity: In general, this paper is well written and easy to follow. But there are some technical points that need to be further clarified as mentioned in the weaknesses.

Relation to Prior Work: Yes.

Reproducibility: Yes

Additional Feedback:

[Author Response · NeurIPS 2020]

**Clarify models and equations**: **(R2)** Eq 1: The recurrent model contains an LSTM and a language graph. The global attended context $\tilde{h}_{t-1}^g$ produced by the language graph is fed to the LSTM as a language-aware "hidden state". The output of LSTM is fed back to the language graph as recurrent hidden state. **(R2)** Eq 1: We have experimented with feeding the object features to the LSTM but the agent performance degenerates. One possible reason is that Eq 1 keeps track of the navigation process and the completion of the instruction, but the same objects can appear in many viewpoints and disturb the tracking. **(R2)** Line 164: Empirically, we found that the temporal-link on the action node has the strongest influence on the agent's performance, compared with having the temporal-link on either the scene or the object node, or on more than one visual node. We believe this is because the temporal-link on the action node explicitly informs the visual graph about the action performed by the agent at the previous step, which could be used by the visual graph to infer a better action at the current step. **(R3)** Eq 10: Following previous work, we trained two models for R2R and R4R tasks respectively, using the same learning rate and optimizer as in the baseline method EnvDrop [28]. The coefficient $\lambda$ is set to 0.2 in both tasks. **(R4)** Eq 2: The same $h_t$ and $u$ are fed into the attention modules with different learned parameters (see Appendix A.1 and [34]). The three attended contexts are used to initialize corresponding visual node features. Hence the three attended contexts will be relevant for three different visual clues.

**Complexity (R1, R3)**: VLN is a complicated task, instead of having a monolithic network [9, 28], we add structure to our model to enhance its learning (better performance) and its interpretability. Introducing such structure/complexity in design actually reduces the complexity in training and inference: concurrent work PREVALENT [9] performs pre-training with 6,582K image-text-action triplets on eight V100 GPUs, but our work does not. Our model has about 22M parameters, which is larger than the baseline model EnvDrop (13M) [28], but much smaller than PREVALENT (190M, roughly LXMERT+EnvDrop). As for inference time, our method is 1.37 times faster on average than EnvDrop on the validation unseen split, because our agent is more efficient in solving the navigation tasks.

**Performance**: **(R1)** In contrast with concurrent work that applies auxiliary reasoning tasks (AuxRN [36]) or performs pretraining (PREVALENT [9]), our method without applying those training techniques still achieves better results in NE, SR and SPL metrics on the R2R test split. **(R3)** The ranking on leaderboard does not distinguish different experiment settings (e.g., single-run, beam-search, pre-exploration or multiple-instructions). Our method achieves the best SPL under the single-run setting (the primary setting for VLN)(see Line 236 and Table 1 caption). **(R4)** We follow the common practice of reporting the best performing model on the validation-unseen split during training [28,26,9] and evaluate it on the test split. We agree with the reviewer to analyze the failure modes, especially given our findings in the Ablation study. We will add some examples of failure cases to the Appendix of our paper. **(R4)** As in some recent work [9,19], we did not evaluate our agent with pre-exploration, which is not an original setting in VLN. However, we agree with R4 that learning a visual graph for the pre-explore environments and incorporate the pre-learned graph into training is a promising direction. We will consider it in our future work.

**Contribution and Theory (R3)**: Based on our observation of the given instructions, in theory, if an agent can identify the three visual clues in the environment and learn about their relationships, it is more likely that it can successfully interpret the complex instructions and correctly perceive the environment. Therefore, we purpose a novel graph network for VLN to capture and utilize the inter- and intra-modal relationships among language and visual entities. We empirically validate our theory by evaluating our method on R2R and R4R benchmarks. We show the importance of the visual clues through our qualitative analysis, and the effectiveness of the two graph networks in the ablation studies.

**Generalizability**: **(R2)** We thank the reviewer's suggestion of evaluating our method on other VLN datasets such as Touchdown (street view) and CVDN (indoor dialog) to show the generalizability. Considering that our proposed relationship graph is a visual-textual encoder, it can be applied on these datasets with minor modifications. We will leave it as further work and add it to our (public) git repository. **(R4)** Our work is tailored for VLN because we want to solve VLN, which is a complicated and significant problem (and application). However, our work is not limited to VLN, e.g., it can be applied to temporal localization with languages in videos with some minor changes.

**Ablation study and Future direction**: **(R1, R4)** The removal of language graph means the message passing in the visual graph is not conditioned on the relational contexts. The removal of visual graph means there is no message passing (hence, the language graph has no effect). In both cases, the action prediction (Eq 9) is conditioned on all the existing visual node features. From model #4, #5 and Full, the object features give a large boost in performance when the two graph networks exist. **(R2)** Line 280: Our observation here was made during preliminary investigations and not part of our main result. However, we believe the finding is important and we would like to share it with other researchers, we will put the result on the final version of our paper or to our git repository. **(R4)** Self-supervised learning: As explained in Line 231-232, we applied the self-supervised learning as in EnvDrop [28], which uses a trained speaker [8,28] to generate instructions for randomly sampled trajectories for data augmentation.

We thank all the reviewers for acknowledging the contribution and strengths of our paper, as well as the thoughtful comments and the constructive suggestions. We will clarify all the unclear points and add the missing reference in the final version of our paper.

[Meta-Review · NeurIPS 2020]

The approach is general which may allow for extensions to incorporate other signals not typically available. This work also provides a mechanism for inspecting the contributions of the factors on language attention. Additionally, the authors use more than one VLN setting to show the generalization. However, the approach is complicated which may limit adoption. Additional analysis to provide an intuitive sense of the approach's strengths and weaknesses will strengthen its place in the literature.